# OxEnsemble: Fair Ensembles for Low-Data Classification

**Jonathan Rystrøm**[1]
[1] *Oxford Internet Institute, University of Oxford, Oxford, UK*

**Zihao Fu**[2]
[2] *The Chinese University of Hong Kong, Hong Kong, China*

**Chris Russell**[1]                    FIRSTNAME.LASTNAME@OII.OX.AC.UK

**Editors:** Accepted for publication at MIDL 2026

## Abstract

We address the problem of fair classification in settings where data is scarce and unbalanced across demographic groups. Such low-data regimes are common in domains like medical imaging, where false negatives can have fatal consequences.

We propose a novel approach *OxEnsemble* for efficiently training ensembles and enforcing fairness in these low-data regimes. Unlike other approaches, we aggregate predictions across ensemble members, each trained to satisfy fairness constraints. By construction, *OxEnsemble* is both data-efficient – carefully reusing held-out data to enforce fairness reliably – and compute-efficient, requiring little more compute than used to fine-tune or evaluate an existing model. We validate this approach with new theoretical guarantees. Experimentally, our approach yields more consistent outcomes and stronger fairness-accuracy trade-offs than existing methods across multiple challenging medical imaging classification datasets.

## 1. Introduction

Deep learning performs exceptionally well when trained on large-scale datasets (Deng et al., 2009; Gao et al., 2020; Hendrycks et al., 2020), but its performance deteriorates in small-data regimes. This is especially problematic for marginalised groups, where labelled examples are both scarce and demographically imbalanced (D'ignazio and Klein, 2023; Larrazabal et al., 2020). In medical imaging, underrepresentation of minority groups leads to poor generalisation and higher uncertainty (Ricci Lara et al., 2023; Mehta et al., 2024; Jiménez-Sánchez et al., 2025). As a result, the very groups most at risk of discrimination are also those for which deep learning methods work least well.

Existing fairness methods often fail in low-data settings (Piffer et al., 2024). As data on disadvantaged groups is needed to learn effective representations *and* to estimate group-specific bias, most methods underperform empirical risk minimisation (Zong et al., 2022).

Ensembles offer a natural way to address these challenges. By aggregating predictions across members, ensembles make more efficient use of scarce examples while leveraging disagreement between members for robustness (Theisen et al., 2023). This makes ensembles particularly attractive for fairness in low-data regimes, but without theoretical foundations, improvements remain inconsistent (Ko et al., 2023; Schweighofer et al., 2025).

We address this by introducing OXENSEMBLE: ensembles explicitly designed to enforce fairness constraints at the member level and provably preserve them at the ensemble level. Our theoretical results show when minimum rate and error-parity constraints are guaranteed

RYSTRØM [ID] FU RUSSELL [ID]  FIRSTNAME.LASTNAME@OII.OX.AC.UK

| Paper | Deep | Img. | Interv. | Min. Rates | Low-data |
|---|---|---|---|---|---|
| Grgić-Hlača et al. (2017) | ✗ | ✗ | ✗ | ✗ | ✗ |
| Bhaskaruni et al. (2019) | ✗ | ✗ | ✓ | ✗ | ✗ |
| Gohar et al. (2023) | ✗ | ✗ | ✗ | ✗ | ✗ |
| Ko et al. (2023) | ✓ | ✓ | ✗ | ✗ | ✗ |
| Claucich et al. (2025) | ✓ | ✓ | ✓ | ✗ | ✗ |
| Schweighofer et al. (2025) | ✓ | ✓ | ✓ | ✗ | ✗ |
| **OxEnsemble** | ✓ | ✓ | ✓ | ✓ | ✓ |

(*a*) Comparison with related work.

(*b*) OxEnsemble pipeline

Figure 1: **(a) Comparisons.** We compare existing works on whether they study deep ensembles; have been applied to images; propose fairness interventions; enforce minimum rates, and target low-data regimes. **(b) OxEnsemble pipeline.** *Train (1):* Members share backbone and task + protected attributes. *Validate (2):* Enforce fairness constraint while maximising accuracy. *Predict (3):* Majority vote. Partitioning ensures full coverage; shared backbone improves efficiency, and voting provides guarantees.

to hold, and how much validation data is required to observe these guarantees in practice. Empirically, we demonstrate that OxEnsemble outperforms strong baselines in medical imaging—where fairness is urgently needed but data for disadvantaged groups is limited.

We make three contributions:

1. **Method:** We introduce an efficient ensemble framework of fair classifiers (OxEnsemble) tailored to fairness in small image datasets.

2. **Theory:** We prove that our fair ensembles are guaranteed to preserve fairness under both error-parity and minimum rate constraints, and we derive how much data is required to observe minimum rate guarantees in practice.

3. **Results:** Across three medical imaging datasets, our method consistently outperforms existing baselines on fairness–accuracy trade-offs.

The article is organised as follows: § 2 presents related work in low-data fairness and fairness in ensembles. § 3 describes both how we construct and train the ensemble (§ 3.1) and the formal guarantees for when it works (§ 3.2). Finally, § 4 and § 5 provide empirical support for the benefits of fair ensembles versus strong baselines on challenging datasets.

## 2. Related Work

**Fairness Challenges in Low-Data Domains:** Deep learning methods achieve near-human performance on overall metrics (Liu et al., 2020), yet consistently underperform for

marginalised groups in medical imaging (Xu et al., 2024; Daneshjou et al., 2022; Seyyed-Kalantari et al., 2021). A central source of bias is unbalanced datasets (Larrazabal et al., 2020), where disadvantaged groups lack examples to learn reliable representations, leading to poor calibration and uncertain predictions (Ricci Lara et al., 2023; Mehta et al., 2024; Christodoulou et al., 2024).

Defining fairness is equally challenging. Standard parity-based metrics such as equal opportunity (Hardt et al., 2016) can be satisfied trivially by constant classifiers in imbalanced datasets and often reduce performance for all groups, a phenomenon of "levelling down" with serious real-world consequences (Zhang et al., 2022; Zietlow et al., 2022; Mittelstadt et al., 2024). In safety-critical domains such as medicine, *minimum rate constraints*—which enforce a performance floor across groups—are often more appropriate to ensure that classifiers serve all subpopulations (Mittelstadt et al., 2024). For further works, see Appendix H.

**Fairness in Ensembles:** Prior work has observed that ensembles sometimes improve fairness by boosting performance on disadvantaged groups (Ko et al., 2023; Schweighofer et al., 2025; Claucich et al., 2025; Grgić-Hlača et al., 2017). However, these studies are observational: improvements are not guaranteed, and in some cases ensembles can even worsen disparities (Schweighofer et al., 2025). Our approach is interventionist. Building on theoretical results for ensemble competence (Theisen et al., 2023), we extend their proofs to fairness settings. This allows us to show formally *why and when* ensembles improve fairness, unlike prior works which only demonstrated that they sometimes do. See Table 1(a) for a complete comparison with related works.

Schweighofer et al. (2025) proposed per-group thresholding (Hardt et al., 2016) to enforce equal opportunity on an ensemble's output. This may not work for imaging tasks as it requires explicit group labels that are not part of images. It is also inappropriate for low-data regimes as it requires a large held-out test set to reliably correct for unfairness.

## 3. Methods

At its heart, this paper looks to circumvent a fundamental trade-off:

*Held-out data must be used to reliably measure and remove bias (Zietlow et al., 2022), but holding back data reduces performance of the base classifier – particularly on data-scarce minority groups.*

We circumvent this trade-off through ensemble-based data reuse. Each member of the ensemble has its fairness enforced using held-out data. However, as this data changes from ensemble member to member, the ensemble as a whole has better generalisation than a single member. Novel theoretical results show that we can expect fairness at the member level to transfer to behaviour of the ensemble as a whole (see § 3.2).

**Choice of fairness constraints:** We focus on two fairness constraints: *equal opportunity* ($EO_p$, the maximum difference in recall across groups; Hardt et al., 2016) and *minimum recall* (the lowest recall of any group ; Mittelstadt et al., 2024). Both target false negatives, which is appropriate when missing a positive case (e.g., a deadly disease) is far more costly than overdiagnosis—a scenario that is especially relevant in medical imaging (Seyyed-Kalantari et al., 2021). Of the two measures, we believe *minimum recall rates* to be more clinically relevant, while *equal opportunity* is more common in the field. While we highlight these

RYSTRØM🆔 FU RUSSELL🆔      FIRSTNAME.LASTNAME@OII.OX.AC.UK

two constraints, our approach can be applied to any other fairness metrics supported by OxonFair (Delaney et al., 2024).

### 3.1. Ensemble Construction and Training

We consider an ensemble of deep neural networks (DNNs) sharing a pretrained convolutional backbone (Figure 1(b)). Each ensemble member is trained on a separate fold, stratified by both target label and group membership (T r et al., 2023). Training each member on different folds allows us to fully utilise the dataset, unlike standard fairness methods that require held-out validation data (Hardt et al., 2016; Delaney et al., 2024). Predictions are aggregated by majority voting, which enforces the guarantees of Theisen et al. (2023) (see § 3.2).

**Enforcing the fairness of ensemble members:** Each ensemble member is trained as a multi-headed classifier following OxonFair (Delaney et al., 2024). These heads predict both the task label (e.g., disease vs. no disease) and the protected attribute (i.e., group membership; see Figure 1(b), left). The task prediction head is trained with standard cross-entropy loss, while the group heads predict a one-hot encoding of the protected attribute using a squared loss.

The two heads are combined using OxonFair's multi-head surgery. This procedure takes a weighted average of the heads and a constant classifier, with weights selected on a validation set to enforce fairness constraints (e.g. the difference in recall between groups is less than 2% or the minimum recall over any group is more than 70%) while maximising accuracy. This averaging process can be performed in place, resulting in a single fair classifier with the same architecture as the single-headed model that predicts the original task label. (See Delaney et al., 2024, section 4.2 for details).

This formulation allows any group fairness definition that can be expressed as a function of per-group confusion matrices to be optimized. Because weights are selected using held-out data, we can enforce error-based criteria—such as equal opportunity or minimum recall—even when the base model overfits during training. In practice, we enforce fairness per member using the held-out data of their corresponding fold, and we optimize over accuracy together with an experiment-specific fairness constraint: either minimum recall or equal opportunity.

**Efficient ensembling of deep networks:** The main computational bottleneck in deep CNNs is the backbone. To avoid repeatedly running the same backbone for ensemble members, we concatenate all classifier heads on a shared backbone. During training, the loss is masked so only the relevant head is updated for each data point. When the backbone is pretrained and frozen,[1] this is equivalent to training each member independently while requiring only a single backbone pass. A related idea with backbone fine-tuning is described by Chen and Shrivastava (2020). We use EfficientNetV2 (Tan and Le, 2021) pretrained on ImageNet (Deng et al., 2009) as the backbone in all experiments. We show alternative, but qualitatively similar, results with MobileNetv3 in Appendix I (Howard et al., 2019).

This yields substantial efficiency gains. Inference speed is essentially identical to a single ERM model, while training is somewhat slower due to multiple heads, but still much faster than training all members separately (which would be about $M\times$ slower for an M-member

---

1. Freezing the backbone helps prevent overfitting on small datasets.

ensemble). Appendix F provides empirical comparisons for the efficiency gains (see Tables 2 and 6), and Appendix A gives implementation details. To ensure robustness, each experiment is repeated over three train/test splits.

## 3.2. Formal Guarantees for Fairness

We now ask: under what conditions can ensembles be expected to *guarantee* fairness improvements? As mentioned in § 2, most prior work on fairness in ensembles is observational, showing that ensembles sometimes improve fairness (Claucich et al., 2025; Ko et al., 2023, e.g.,), while Schweighofer et al. (2025) showed that fairness could be enforced on the output of an ensemble using standard postprocessing. In contrast, we take an interventionist approach and ask, *after enforcing fairness per ensemble member, can we expect it to transfer to the ensemble as a whole?*. We provide theoretical conditions under which fairness is improved,and show how it can be used in practice.

The theory is based on Theisen et al. (2023), who show that *competent* ensembles never hurt accuracy. Informally, an ensemble is competent over a distribution $D$ if it is more likely to be confidently right than confidently wrong. Let the error rate of an ensemble $\rho$ be:[2]

$$W_\rho = W_\rho(X, y) = \mathbb{E}_{h \sim \rho}[1(h(X) \neq y)]$$

and define

$$C_\rho(t) = P_{(X,y) \sim D}(W_\rho \in [t, 1/2)) - P(W_\rho \in [1/2, 1-t]) \ \forall t \in [0, 1/2)$$

The ensemble is *competent* if $C_\rho(t) \geq 0$ for all $t \in [0, 1/2)$. This definition makes no distributional assumptions and can be verified on held-out data.

Theisen et al. (2023) showed that if competence holds on a dataset $(X, y)$, then majority voting improves accuracy relative to a single classifier, with the improvement bounded by the disagreement between members.

To extend competence to fairness metrics, we evaluate competence on *restricted subsets of the data.* Let $\mathcal{G}$ be the set of protected groups. For any group $g \in \mathcal{G}$, write $g+$ for the positives $(y = 1, A = g)$ belonging to a group. We similarly write $D^+$ for the set of all positives in the distribution.We define

$$C_\rho^{g+}(t) = P_{(X,y) \sim g+}(W\rho \in [t, 1/2)) - P_{(X,y) \sim g+}(W\rho \in [1/2, 1-t]) \tag{1}$$

We say an ensemble is *restricted groupwise competent* if $C_\rho^{g+}(t) > 0$ for all $t, g \in G$, and say it is *restricted competent* if $C_\rho^{D^+}(t) > 0$.

Based on this, we derive three main results:

1. **Minimum rate constraints:** If an ensemble is restricted groupwise competent, and every member of the ensemble satisfies a minimum rate constraint, then the ensemble as a whole also satisfies that minimum rate.

2. **Error parity:** If an ensemble is restricted groupwise competent, and if every member of the ensemble approximately satisfies an error parity measure (e.g., equal opportunity), then the ensemble as a whole also approximately satisfies it. The achievable bounds depend on disagreement- and error rates of the members.

---

2. For definitions of all notation used see Table 5.

RYSTRØM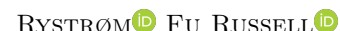 FU RUSSELL      FIRSTNAME.LASTNAME@OII.OX.AC.UK

3. **Restricted Groupwise Competence** can be enforced by appropriate minimum recall constraints.

Together these results show how ensemble competence on restricted subsets provides guarantees for both minimum rate constraints and error parity measures, covering a broad range of fairness definitions. Moreover, (iii) shows that the conditions required for the theorem to hold are exactly those enforced by setting minimum recall rates.

We begin with a lemma.

**Lemma 1** *Restricted competent ensembles do not degrade recall relative to the average recall of a member.*

**Proof** Proof follows immediately by applying the main result of Theisen et al. (2023) to $D^+$ rather than $D$, and observing that accuracy when restricted to the positives is equivalent to recall.[3]

This main result bounds the *Error Improvement Rate (EIR)*—the ensemble's relative improvement over a single classifier—by the *Disagreement Error Ratio (DER)*. See Appendix C for formal definitions. For binary classification, the bounds are given by Eq. 2 for an arbitrary data distribution, $\mathcal{D}$:

$$\text{DER}_\mathcal{D} \geq \text{EIR}_\mathcal{D} \geq \max(\text{DER}_\mathcal{D} - 1, 0) \tag{2}$$

Replacing $D$ with $D^+$ implies the error improvement rate on the positives must be non-negative for a *restricted competent* ensemble as required. ∎

### 3.2.1. RESTRICTED GROUPWISE COMPETENCE GUARANTEES

**1. Minimum rates for competent ensembles:** We apply the result from lemma 1 to each group independently. We observe that if the ensemble is *restricted groupwise competent*, the recall rate for each group can not degrade by ensembling. Therefore the minimum recall rate for any group, must also not be degraded. ∎

**2. Error parity from competence:** Error-parity constraints such as approximate equal opportunity (equality of recall across groups; Hardt et al., 2016) or approximate equality of accuracy (Zafar et al., 2019) are harder to guarantee. The difficulty is that while ensembles can improve average performance, unequal improvements across groups can increase disparities (see, e.g., Schweighofer et al., 2025). Nonetheless, *restricted groupwise competence* still yields limited but useful bounds.

We consider the $L_\infty$ form of approximate fairness: a classifier has $k$-approximate fairness with respect to groups $\mathcal{G}$ if

$$k \geq \max_{g \in \mathcal{G}} L_g(h) - \min_{g \in \mathcal{G}} L_g(h) \tag{3}$$

where $L_g$ is the average loss on group $g$, corresponding to 1 minus one of the measures we are concerned with (typically recall). The question then is, if every member of the ensemble exhibits $k$-approximate fairness, what fairness bounds do we have for the ensemble?

---

3. A similar argument can be made using the negatives and *sensitivity*.

By applying Eq. 2 (see Appendix G.3 for derivation), we obtain the following bound:

$$k^* \leq k + \max_{g \in \mathcal{G}} \mathbb{E}_{h \sim \rho}[L_g(h)]\text{DER}_{g^*} - \max(0, \min_{g \in \mathcal{G}} \mathbb{E}_{h \sim \rho}[L_g(h)](\text{DER}_{g^*} - 1)) \quad (4)$$

Both bounds are pessimistic. In practice, our approach works well for enforcing equal opportunity (see § 5). Still, two insights follow: First, viewed through the governance lens of *levelling down* (Mittelstadt et al., 2024) these fairness violations are less concerning. Fairness was enforced per ensemble member, and presumably performance per group was set at an acceptable level. Any subsequent unfairness comes because groups are doing better than expected, rather than worse. Second, the bound scales with $L_g$, and therefore the worst-case disparity shrinks as group losses decrease. In practice, this means that enforcing additional minimum rate constraints through our method can tighten the bounds.

### 3.2.2. Guarantees for Minimum Recall

The previous section showed that restricted groupwise-competent ensembles can improve minimum rates and fairness. In this section, we show how to ensure restricted groupwise competence by setting minimum recall rates.

Enforcing minimal recall rates for each ensemble member alters the decisions made. Looking at Eq. 1, we observe that increasing the recall rate for all ensemble members over some group $g$ decreases the probability of error over the positives. As such, enforcing a sufficiently high recall rate can guarantee competence (i.e., perfect recall implies no errors and therefore competence).

In practice, identifying the smallest minimal recall rate that guarantees competence is an empirical question and requires a further held-out set to measure competence as a function of minimal recall. Given the paucity of data, we are unable to do this. Instead, we prove that, for a minimum recall rate of more than 50%, competence is guaranteed for an ensemble where the members make independent errors. See appendix G.1 for details. This result is consistent with *Jury Theorems* (Condorcet, 1785; Berend and Paroush, 1998; Kanazawa, 1998; Pivato, 2017) that show that majority votes from mildly correlated voters with average accuracy $> 0.5$ improve over individual voters, converging to perfect accuracy as ensemble size increases (Mattei and Garreau, 2025). We emphasise that only the specific value of 0.5 depends on independence assumptions. The existence of *some* threshold does not, and neither does the rest of the theory.

Similarly, when the minimum recall for every member falls below (50%), independent ensembles are not restricted groupwise competent. We demonstrate this also holds empirically in Fig. 2, where no group achieves competence when $k < 0.5$ across two datasets (see § 4).

### 3.2.3. Minimum validation and evaluation sizes

Under the assumption of independent errors, a minimum recall of $k > 0.5$ on the test set, guarantees that the ensemble will also have a minimum recall of $k$. The challenge here is that recall constraints are imposed on validation data, and as we are dealing with very low-data groups, sometimes with $< 100$ positive cases, the constraints need not generalise to test data.

To ensure these constraints generalise to test data, we want to determine the minimum recall, $P_{\min}$, required the on a validation set with $m$ positives in the minority group such

RYSTRØM [ID] FU RUSSELL [ID]          FIRSTNAME.LASTNAME@OII.OX.AC.UK

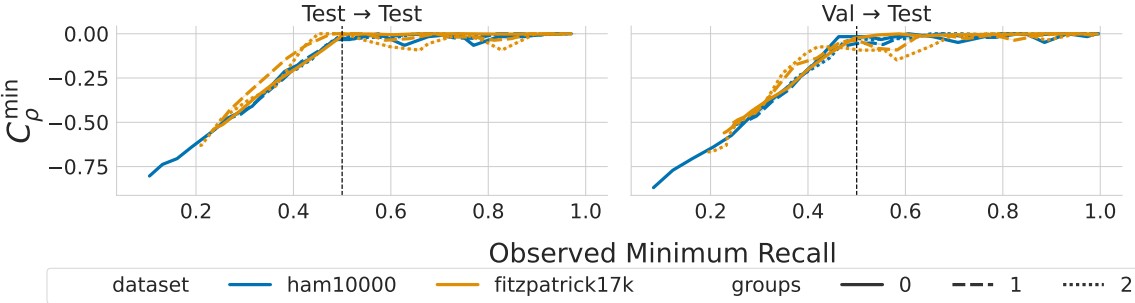

Figure 2: **Competence Violations vs Recall.** Competence violations ($C_\rho$; 0=perfect) are high when recall<0.5 and stabilize at recall>0.5. *Left*: Test set for fitting and evaluation. *Right*: Validation set for fitting, test set for evaluation.

that with a probability $\alpha$, the recall on an evaluation set with $n$ positives will be at least 50%. This guarantees that the minimum recall of the ensemble is greater than the average recall of each member.

We assume that validation and test sets are of known sizes, $m$ and $n$ respectively, and drawn from the same distribution. By drawing on the literature for one-sided hypothesis tests on Bernoulli distributions, we arrive at Eq. 5.

$$p_{\min} = 0.5 + \tfrac{1}{2} z_{1-\alpha} \sqrt{\tfrac{1}{m} + \tfrac{1}{n}}. \tag{5}$$

Here $z_{1-\alpha}$ is the z-score for significance $1 - \alpha$. The primary implication of Eq. 5 is that larger $n$ decreases the need for high validation thresholds – especially in small-data settings. For derivations see Appendix G.2. We find empirical support for our theoretical guarantees of fairness on positive samples in Appendix E. Here, we show that as long as the minimum recall is enforced at a sufficiently high threshold, we observe restricted groupwise competence on the test set.

This result is more generally applicable outside of fairness, and to ensure a classifier has a recall of more than $k$ with probability $\alpha$, on an unseen test set, the recall on a validation set should be set to more than

$$p_{\min} = k + z_{1-\alpha} \sqrt{k(1-k)\left(\tfrac{1}{m} + \tfrac{1}{n}\right)}.$$

See Appendix G.2 for more details.

## 4. Experimental Setup

**Data and Protected Attributes**   We evaluate on three medical imaging datasets from MedFair (Zong et al., 2022) and FairMedFM (Jin et al., 2024)—see Table 1. Each task is a binary classification with image-only inputs (discarding all auxiliary features for fair comparison). For Fitzpatrick17k, the common binary split (I–III vs. IV–VI) can mask harms to the darkest tone (VI), which comprises only 0.4% of positives. We instead separate out V and VI, grouping I–IV to preserve adequate support elsewhere.

Table 1: Evaluation datasets. "Min. Positives" is the number of *positive* examples in the smallest group (bold). These small counts stress-test low-data fairness.

| Dataset | Task | # Min. Positives | Protected Attributes |
|---|---|---|---|
| HAM10000 | Skin cancer | 94 | Age (0-40, **40-60**, 60+) |
| Fitzpatrick17k | Dermatology | 60 | Skin type (I-IV, V, **VI**) |
| Harvard-FairVLMed | Glaucoma | 399 | Race (Asian, White, **Black**) |

**Preprocessing and splits:** Images are centre-cropped and resized to 224x224 (Deng et al., 2009) with random augmentations during training. Dataset-specific validation/test sizes follow § 3.2.3 to guarantee 70% minimum observable recall. See Appendix A for full details.

**Evaluation Metrics:** Medical classification is a non-zero-sum game where "levelling down"—reducing groups' performance to achieve parity—can have fatal consequences (Mittelstadt et al., 2024). The predominant harm is failing to diagnose ill people from disadvantaged groups, making *minimum recall* a more appropriate metric than disparity-based measures such as equal opportunity. Moreover, with positive class incidence below 10% for disadvantaged groups, a trivial all-negative classifier achieves high accuracy, and perfectly satisfies equal opportunity, while misclassifying all sick patients.

However, a key question when using *minimum recall rates* is "What should the rate be set to?" Our position is that this is a deployment decision that must be made on a case-by-case basis. As such, our primary metric, FairAUC, summarizes the possible choices by averaging the best accuracy $a$ achievable for each minimum recall threshold $t \in T$:

$$\text{FairAUC} = \frac{1}{|T|} \sum_{t \in T} \left( \max_{(a,r) \in M, r \geq t} a \right) \quad (6)$$

where $M$ are model configurations and $r$ is minimum recall. We evaluate over $T \in [0.5, 1]$—the zone with theoretical guarantees (§ 3.2). Confidence intervals use 200 bootstrap samples at 95%. For baselines without explicit thresholding, we generate Pareto frontiers by varying global thresholds on validation data.

**Baselines and Ensemble Settings:** We compare against established fairness methods to ensure a meaningful contribution. As a reference, **Empirical Risk Minimisation (ERM)** minimises training error without considering fairness (Vapnik, 2000). We include **Domain-Independent Learning**, which trains a separate classifier for each protected class with a shared backbone, and **Domain-Discriminative Learning**, which encodes protected attributes during training and removes them at inference (Wang et al., 2020). **Fairret** introduces a regularisation term accounting for the protected attribute and fairness criterion (Buyl et al., 2023), while **OxonFair** tunes decision thresholds on validation data to enforce group-level fairness (Delaney et al., 2024). **Ensemble (HPP)** implements a homogenous ensemble (similar to Ko et al., 2023) followed by Hardt Post Processing (Hardt et al., 2016) as proposed by Schweighofer et al. (2025). Finally, **Ensemble (ERM)** is equivalent to our method without enforced constraints, serving as an ablation to assess whether the added fairness interventions of OxEnsemble increases FairAUC.

All baselines are trained with the same configuration as our ensembles. Minority groups are rebalanced via upsampling as suggested by Claucich et al. (2025), and we reimplement

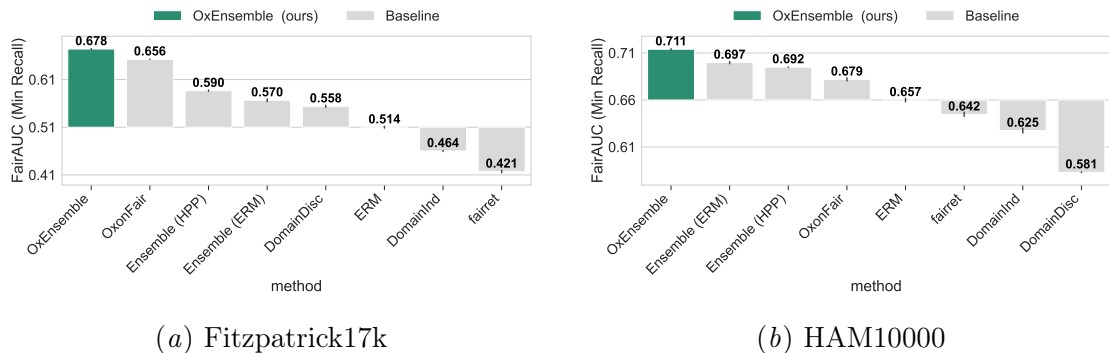

(*a*) Fitzpatrick17k                    (*b*) HAM10000

Figure 3: **Fairness–accuracy AUC (FairAUC) relative to ERM.** OxEnsemble achieves higher FairAUC than all baselines on Fitzpatrick17k (left) and HAM10000 (right). Error bars show 95% bootstrap CIs. Evaluation follows § 4 over minimum-recall thresholds in [0.5, 1].

Table 2: Single-image inference. Details in Appendix F.

| Method | Latency (ms) ↓ | |
| --- | --- | --- |
| | CPU | CUDA |
| ERM | $112.22 \pm 13.58$ | $5.42 \pm 0.31$ |
| Ensemble | $107.15 \pm 12.41$ | $5.83 \pm 0.38$ |

methods following Zong et al. (2022) and Delaney et al. (2024). Fairret requires a hyperparameter search over regularisation weights. To generate comparable Pareto frontiers, we fit global prediction thresholds so that a minimum recall of $k$ is enforced on a held-out validation set, mirroring deployment where thresholds are tuned on available data but applied to unseen test data (Kamiran et al., 2013). For minimum recall experiments, for all methods that do not naturally support minimum recall rates, we select a global threshold that maximises accuracy while achieving a recall $> k$ on the validation set.

**Ensemble size:**  We use 21 members for all ensembles. Appendix D shows that FairAUC is stable across different sizes from 3 to 21 within confidence intervals. We default to 21: it is consistent with our theory that majority voting benefits from more members, while our shared-backbone design keeps inference time essentially unchanged (see Table 2 for efficiency comparisons).

## 5. Results

See Appendix I for similar results with an alternative backbone.

**FairVLMed:**  In Figure 4 (right), only OxEnsemble maintains fairness at strict thresholds (EqualOpportunity $< 4\%$). Most methods break down above 6%. Compared to OxonFair, OxEnsemble achieves higher accuracy with lower fairness violations (Table 3). While standard ensembles have slightly higher accuracy, OxEnsemble consistently reduces disparities

Table 3: Accuracy and fairness violations. Best value in **bold**.

| Dataset | Accuracy ↑ | | Fairness Violations ↓ | |
|---|---|---|---|---|
| | OxEnsemble | OxonFair | OxEnsemble | OxonFair |
| FairVLMed | **0.665** | 0.657 | **0.009** | 0.011 |
| Fitzpatrick17K | **0.642** | 0.623 | 0.057 | **0.048** |
| HAM10000 | **0.707** | 0.679 | **0.067** | 0.082 |

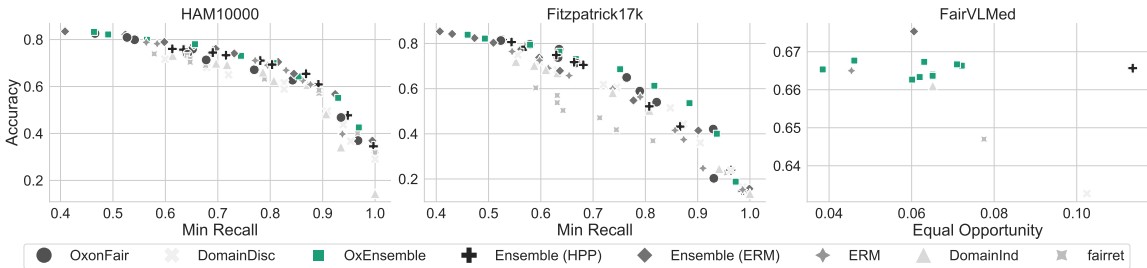

Figure 4: **Pareto frontiers across datasets.** OxEnsemble (green) yields better fairness–accuracy trade-offs than baselines (grey). Left/centre: min recall (HAM10000, Fitzpatrick17k). Right: equal opportunity (FairVLMed). See § 4 for definitions.

further (e.g., equal opportunity from 6% to < 5% with < 1pp accuracy loss). The HPP-based method from Schweighofer et al. (2025) fails to enforce equal opportunity ($EO_p > 11\%$).

**Fitzpatrick17k:** Here, in the most challenging setting (60 positive samples in the smallest group), OxEnsemble clearly outperforms all baselines. It reaches FairAUC = 67.7%, compared to 57.0% for standard ensembles (58.9% with HPP) and 51.3% for ERM (Figure 3(a)). Across thresholds, OxEnsemble is Pareto-optimal (Figure 4, centre).

**HAM10000:** OxEnsemble achieves the highest accuracy and lowest fairness violations. Its FairAUC = 71.1% significantly outperforms ERM (65.7%), baseline ensembles (69.8% & 69.2%), and OxonFair (67.9%). All other methods perform worse than ERM.

## 6. Conclusion

A lack of data for minority groups remains one of the fundamental challenges in ensuring equitable outcomes for disadvantaged groups. We present a novel framework for constructing efficient ensembles of fair classifiers that address the challenge of enforcing fairness in these low-data settings. Across three medical imaging datasets, our method consistently outperforms existing fairness interventions on fairness-accuracy trade-offs. Unlike prior work on ensembles that observed occasional fairness improvements, our approach guarantees that fairness is not degraded and shows that ensembles are a practical tool for reusing scarce data to produce more reliable fairness estimates.

Our theoretical analysis explains *why* these improvements occur. We prove that enforcing minimum rate constraints above 0.5 ensures ensemble competence for the worst-performing groups, derive bounds for error-parity measures such as equal opportunity, and provide

Rystrøm 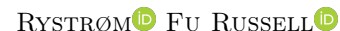 Fu Russell          firstname.lastname@oii.ox.ac.uk

principled guidance on the validation and test sizes needed for these guarantees to hold in practice. Together, these results expand the understanding of both when and why ensembles improve fairness, offering a principled and empirically validated method for building more equitable classifiers in high-stakes domains. Code can be found on GitHub.

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

## Appendix A. Implementation Details

The code and instructions for reproducing the results can be found in our GitHub repository[4]. Optimisation for all models is done using Adam (Kingma and Ba, 2015) with a learning rate of 0.0001.

The test splits for the baseline methods (see § 4) used the same seed as the first ensemble run. All experiments were run with deterministic seeds for reproducibility (see repository).

To choose the sizes of the validation and test sets, we use the theory described in § 3.2.3. Applying a minimum observable recall of 70%, we obtain the following sizes. These were applied consistently across all methods.

- **Fitzpatrick17K**: $|\mathcal{D}_{\text{valid}}| = 33\%, |\mathcal{D}_{\text{test}}| = 25\%$

- **HAM10000**: $|\mathcal{D}_{\text{valid}}| = 20\%, |\mathcal{D}_{\text{test}}| = 20\%$

- **FairVLMed**: $|\mathcal{D}_{\text{valid}}| = 10\%, |\mathcal{D}_{\text{test}}| = 10\%$

---

4. Link: https://github.com/jhrystrom/guaranteed-fair-ensemble

For fairret, we evaluate a set of regularisation parameters ranging from 0.5 to 1.5, including [0.5, 0.75, 1.0, 1.25, 1.5]. While Buyl et al. (2023) technically doesn't require a validation set, it makes use of a hyperparameter to govern the fairness/accuracy trade-off. This hyperparameter cannot be set a priori, and must be tuned for every dataset, requiring the use of validation data. We do not conduct any additional parameter search for Domain Discriminative, ERM, or Domain Independent.

All training was done on a single H100. For the final results of the paper, we ran analysis on 3 datasets for 3 iterations using Weights & Biases (Biewald, 2020). Each run took ~11 minutes. In addition, the baseline experiments add an extra 20 runs. In total, this results in approximately 14.5 hours of compute to reproduce the complete results. Note that the experiments could have been run on cheaper hardware since the EfficientNetV2 models only have 43M parameters.

While the above details the compute used to produce the results from the paper, we conducted further experiments before this. Particularly, we experimented with a less efficient ensemble structure requiring a separate run for each ensemble member. This required significantly more compute time.

## Appendix B. Data Access and Information

We provide links for accessing the data in Table 4. While all data is openly available for academic research, some of it requires approval by the providers.

For detailed summary statistics for HAM10000 and Fitzpatrick17k, see the supplemental material in MedFair (Zong et al., 2022). For FairVLMed, we refer to the FairCLIP paper (Luo et al., 2024) as well as the GitHub page. For further details, see the original publications.

Table 4: Dataset access information

| Dataset | Access URL | Reference |
|---|---|---|
| Fitzpatrick17k | https://github.com/ mattgroh/fitzpatrick17k | (Groh et al., 2021) |
| HAM10000 | https://dataverse. harvard.edu/dataset. xhtml?persistentId=doi: 10.7910/DVN/DBW86T | (Tschandl et al., 2018) |
| FairVLMed | https://github.com/ Harvard-Ophthalmology-AI-Lab/ FairCLIP | (Luo et al., 2024) |

## Appendix C. Theoretical formalisms

Table 5 defines all notation used in the main paper.

As mentioned in the main paper, (Theisen et al., 2023) bound the improvements of an ensemble (i.e., the *Ensemble Improvement Ratio (EIR)*) by the *Disagreement-Error Ratio*

Table 5: Summary of notation used in § 3.2.

| Symbol | Definition |
|---|---|
| $\mathcal{D}$ | Data distribution over $(X, Y)$ |
| $X$ | Input features |
| $Y \in \{0, 1\}$ | Binary label ($1$ = positive, $0$ = negative) |
| $A \in \mathcal{G}$ | Protected attribute; $\mathcal{G}$ is the set of groups |
| $g \in \mathcal{G}$ | A particular protected group |
| $\mathcal{D}_{g,+}, \mathcal{D}_{g,-}$ | Conditional distributions $\mathcal{D}\,\vert\,(A = g, Y = 1)$ and $\mathcal{D}\,\vert\,(A = g, Y = 0)$ |
| $g+, g-$ | Shorthand for positives ($A = g, Y = 1$) and negatives ($A = g, Y = 0$) |
| $h$ | Individual classifier (ensemble member) |
| $h'$ | Another (distinct) ensemble member |
| $\rho$ | Distribution over ensemble members (uniform in practice) |
| $h_{\mathrm{MV}}$ | Majority-vote classifier induced by $\rho$ |
| $N$ | Ensemble size (number of members) |
| $L_{\mathcal{D}}(h)$ | Error rate (0–1 loss) of $h$ on $\mathcal{D}$ |
| $L_g(h)$ | Groupwise loss on group $g$ (e.g., $1 -$ recall or $1 -$ accuracy) |
| $D_{\mathcal{D}}(h, h')$ | Disagreement rate between $h$ and $h'$ on $\mathcal{D}$ |
| $W_\rho(X, Y)$ | Ensemble error rate on $\mathcal{D}$: $\mathbb{E}_{h\sim\rho}[\mathbf{1}\{h(X) \neq Y\}]$ |
| $W_\rho^{g+}$ | Ensemble error rate on positives in group $g$ (i.e., on $\mathcal{D}_{g,+}$) |
| $W_\rho^{g-}$ | Ensemble error rate on negatives in group $g$ (i.e., on $\mathcal{D}_{g,-}$) |
| $t \in [0, 1/2]$ | Margin parameter in competence definitions |
| $C_\rho$ | Competence on $\mathcal{D}$: $P(W_\rho \in [t, 1/2)) - P(W_\rho \in [1/2, 1 - t])$ |
| $C_\rho^{g+}$ | Restricted groupwise competence on $g+$ (analogously $C_\rho^{g-}$ for $g-$) |
| $\mathrm{EIR}_{\mathcal{D}}$ | Error Improvement Rate: $\frac{\mathbb{E}_{h\sim\rho}[L_{\mathcal{D}}(h)] - L_{\mathcal{D}}(h_{\mathrm{MV}})}{\mathbb{E}_{h\sim\rho}[L_{\mathcal{D}}(h)]}$ |
| $\mathrm{DER}_{\mathcal{D}}$ | Disagreement–Error Ratio: $\frac{\mathbb{E}_{h,h'\sim\rho}[D_{\mathcal{D}}(h,h')]}{\mathbb{E}_{h\sim\rho}[L_{\mathcal{D}}(h)]}$ |
| $g^*$ | Index for the distribution on which DER/EIR are computed (e.g., $g+$, $g-$, or full) |
| $k$ | Minimum rate constraint (e.g., minimum recall/sensitivity) |
| $k^*$ | Upper bound on ensemble fairness gap under error-parity bounds |
| $K$ | Number of positive predictions among $N$ members for a datapoint |
| $K_i$ | Bernoulli indicator of the $i$-th member's positive prediction |
| $p_i$ | Success prob. of $K_i$; $p_i = k + \delta$ under enforced minimum rate |
| $\bar{p}$ | Mean recall across members: $\bar{p} = \frac{1}{N}\sum_{i=1}^{N} p_i$ |
| $\delta \geq 0$ | Margin by which enforced minimum rate exceeds $k$ on validation |
| $m, n$ | # positives in validation/test for the minority group (for power analysis) |
| $\alpha$ | Significance level in the one-sided test |
| $z_{1-\alpha}$ | $(1 - \alpha)$-quantile of the standard normal distribution |
| $p_{\min}$ | Minimum observed validation recall to ensure test-time recall $> 0.5$: $p_{\min} = 0.5 + \frac{1}{2}z_{1-\alpha}\sqrt{\frac{1}{m} + \frac{1}{n}}$ |

*(DER)* of the ensemble, i.e., the ratio of the average pairwise disagreement rate to the average error of ensemble members.

For completeness, we repeat their major results below. Note that while (Theisen et al., 2023) considers a fixed distribution $\mathcal{D} = (X, Y)$, which they frequently drop from their notation, we preserve it as we will want to vary $\mathcal{D}$.

Their results are as follows:

The ensemble improvement rate is defined as:

$$\text{EIR}_{\mathcal{D}} = \frac{\mathbb{E}_{h \sim \rho}[L_{\mathcal{D}}(h)] - L_{\mathcal{D}}(h_{\text{MV}})}{\mathbb{E}_{h \sim \rho}[L_{\mathcal{D}}(h)]}. \tag{7}$$

and the Disagreement-Error Ratio as:

$$\text{DER}_{\mathcal{D}} = \frac{\mathbb{E}_{h,h' \sim \rho}[D_{\mathcal{D}}(h, h')]}{\mathbb{E}_{h \sim \rho}[L_{\mathcal{D}}(h)]}. \tag{8}$$

Where $L_{\mathcal{D}}(h)$ is the error rate for classifier, $h$, on data distribution, $\mathcal{D}$, $h_{\text{MV}}$ is the majority vote classifier, $\mathbb{E}_{h \sim \rho}$ indicates the expected value over all ensemble members, and $D_{\mathcal{D}}(h, h')$ is the disagreement rate between classifiers, $h$ and $h'$.

Specifically, the authors provide upper and lower bounds on the EIR. Crucially, this rests on an assumption of *competence*, which informally states that ensembles should always be at least as good as the average member. More formally, (Theisen et al., 2023) state:

**Assumption 1 (Competence)** *Let* $W_{\rho,\mathcal{D}} \equiv W_{\rho}(X, Y) = \mathbb{E}_{h \sim \rho, \mathcal{D}}[\mathbf{1}(h(X) \neq Y)]$. *The ensemble* $\rho$ *is* competent *if for every* $0 \leq t \leq 1/2$,

$$\mathbb{P}(W_{\rho,\mathcal{D}} \in [t, 1/2)) \geq \mathbb{P}(W_{\rho,\mathcal{D}} \in [1/2, 1 - t]). \tag{9}$$

This assumption can be interpreted as formalising the statement that a majority voting ensemble is more likely to be confidently right than confidently wrong.

Based on this assumption, (Theisen et al., 2023) prove the following theorem:

**Theorem 2** *Competent ensembles never hurt performance, i.e., EIR $\geq 0$.*

This assumption is only required to rule out pathological cases. For most real-world examples, this will be trivially satisfied. In the case of binary classification, the bounds on EIR can be simplified to Eq. 2 from the main text.

## Appendix D. Ablation: Ensemble Sizes

In this section, we ask: "How does ensemble size affect performance?" We examine how FairAUC varies with ensemble size on the test set, and whether validation performance predicts test performance.

Our design makes this straightforward: because ensemble members are trained independently, we can form smaller ensembles by subsampling members. We construct ensembles of size $m \in \{3, 5, \ldots, M\}$ with $M = 21$, and compute FairAUC on both validation and test sets for HAM10000 (Tschandl et al., 2018) and Fitzpatrick17k (Groh et al., 2021) across all train/test partitions.

Figure 5 shows no consistent trend: confidence intervals are wide, and performance does not vary systematically with ensemble size. An alternative heuristic is to use validation FairAUC to select ensemble size, but as Figure 6 shows, the relationship between validation and test performance is too noisy to be useful. This is expected, as our method already leverages all non-test data to fit fairness weights.

Lacking a strong empirical heuristic, we adopt the largest ensemble ($M = 21$), which best aligns with our theoretical results: larger ensembles provide stronger guarantees under Jury-theorem arguments (see § 3.2.2).

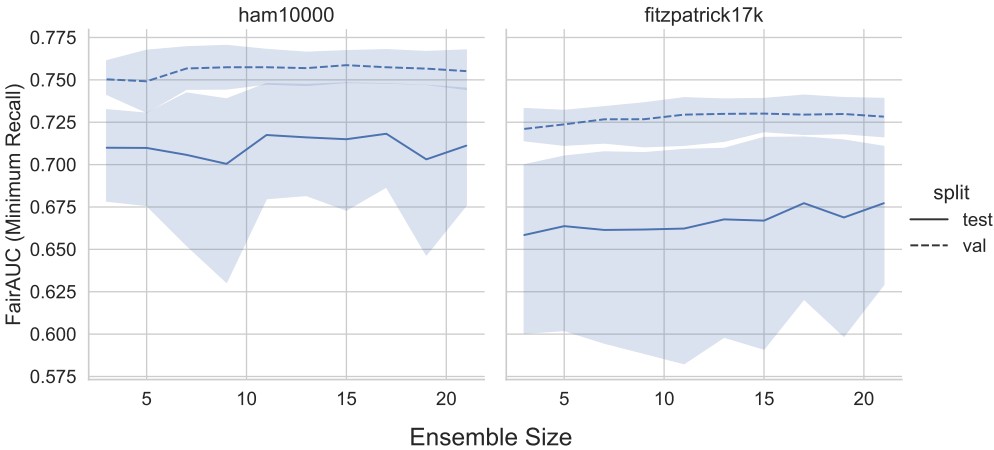

Figure 5: Relationship between **Ensemble Size** (X-axis) and **FairAUC** (Y-axis) across two datasets. No significant relationship is observed.

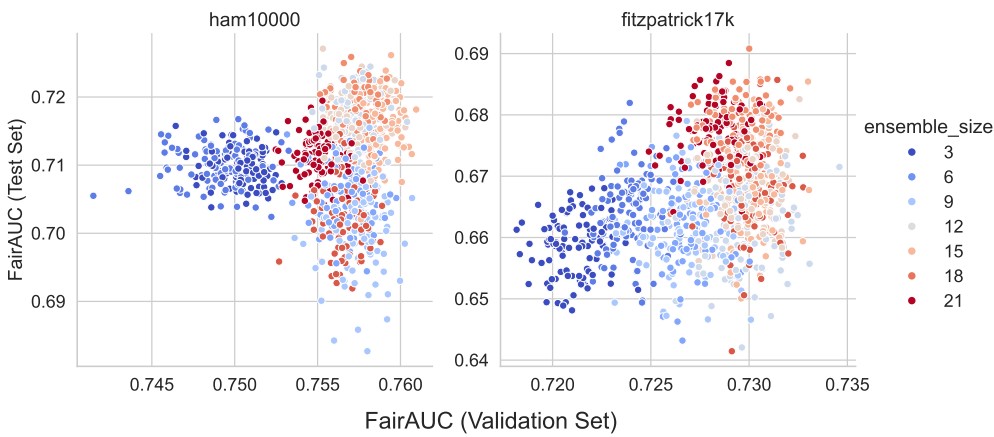

Figure 6: Relationship between FairAUC on validation (X-axis) and test set (Y-axis) across ensemble sizes. The relationship is too noisy to guide model selection.

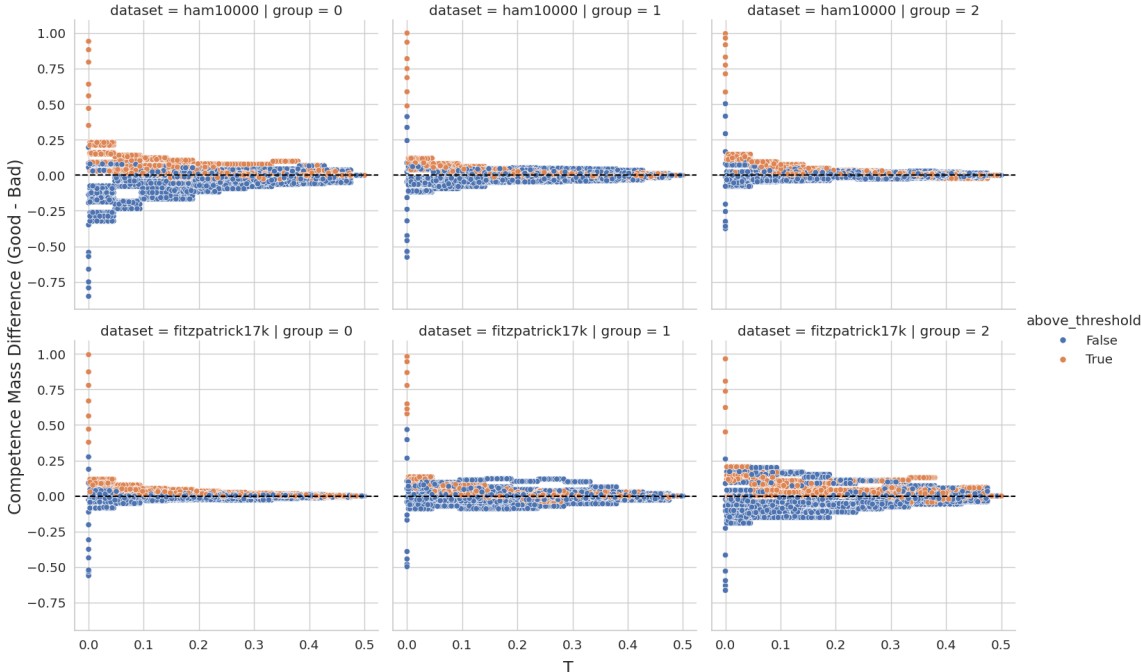

Figure 7: **Empirical validation of competence proofs**. We show that enforcing minimum recall, $k > 0.5 + \delta$, leads to *competent* ensembles (see § 3.2). $\delta$ depends on the data size (§ 3.2.3) and 0.5 comes from our proof in § 3.2.2. The data points *above* thresholds, are above the X-axis, whereas the points *below* the threshold are on both sides.

## Appendix E. Empirical Validation of Competence

We empirically validate our proofs from § 3.2.3 and § 3.2.2. Specifically, we want to show that enforcing recall at $k > 0.5 + \delta$ leads to competent ensembles if $\delta$ is matches the size of the datasets. This would help validate both theoretical extensions of Theisen et al. (2023).

To conduct this analysis, we set threshold $= k + \delta = 0.7$ (as described in Appendix A). We then run the competence calculations from Theisen et al. (2023) for different $k$ above and below the threshold. The resulting figure is Figure 7.

## Appendix F. Benchmarking Efficiency

A big advantage of our OxEnsemble method is that it is efficient for training and inference because it utilises a shared backbone (see § 3.1). In this section, we provide evidence for these claims.

The results for inference can be seen in Table 2. Here, we see comparable inference speeds for ERM and ensemble across both CPU and GPU. The GPU runs are done on an NVIDIA H100 80GB GPU. The runs are with a batch size of 1, averaged over 100 runs, with a warm-up size of 10. There are no significant differences between the methods.

The results for training can be seen in Table 6 based on Weights & Biases data (Biewald, 2020). Here, we see a larger difference; ensembles take approximately 3x longer to train compared to ERM. This may be because we are in essence training 84 times more classifiers (21 members with four heads each). Still, because of the small size of the datasets, the training times are manageable.

It is worth noting that substantial optimisation is available for training. Because the backbone is frozen, the entire evaluation set (validation sets + test set) can be pre-computed. This would drastically speed up the training. However, these optimisations were not done in the interest of time.

Table 6: Average training runtime (in minutes)

| Training Method | Avg. Runtime (min) | Std. Dev. (min) |
| --- | --- | --- |
| Ensemble | 31.79 | 5.13 |
| ERM | 8.51 | 2.28 |

## Appendix G.  Derivations

### G.1.  Restricted Groupwise Competence under minimum recall and Independence Assumptions

To prove this, we assume independence of classifier errors and define $I_p$ for any subset of classifiers $p \in \rho$:

$$I_p(x) = \Pi_{i \in p} P(c_i(x) = 1) \Pi_{j \in \bar{p}} P(c_j(x) = 0) \tag{10}$$

then we decompose

$$P(W_\rho^{g+} = t) = \sum_{\substack{p \in \rho \\ |p| = s}} I_p \tag{11}$$

**Sketch of the proof:**   The proof requires two observations:

1. Negative flips decrease probabilities(given by Lemma 3) Given a subset $p$ of ensemble models taking positive labels, with their complement taking negative labels, flipping some of $p$ so they also take negative labels to obtain a new $q$ subset will result in $q$ having a lower probability of occurring than $p$.

2. Matching $p$s and $q$s(given by Lemma 4) It is possible to identify matching pairs of such $p$ of size $s$ and $q$ of size $|\rho| - s$ in equation Eq. 17 determine.

**Lemma 3**   *If $p \supseteq q$, the following inequality holds for their associated summands:*

$$I_p \geq I_q \tag{12}$$

**Proof**   To see this, we write $n = \bar{p}$ for the members of the ensemble that take a negative label in both $p$ and $q$ and $a = p/q$ for members of the ensemble that alter from positive label to negative as we move from $p$ to $q$. Then

$$\Pi_a(c_a(X) = 1) \geq k^{|a|} \geq (1 - k)^{|a|} \geq \Pi_a(c_a(X) = 0) \tag{13}$$

and

$$\Pi_a(c_a(X) = 1)\Pi_p P(c_p(X) = 1)\Pi_n P(c_n(X) = 0) \geq$$
$$\Pi_a(c_a(X) = 0)\Pi_p P(c_p(X) = 1)\Pi_n P(c_n(X) = 0) \tag{14}$$

As required. ∎

**Lemma 4** *Now we need to establish the existence of a monotonic bijection m that maps from sets of size s to sets of size $|\rho| - s$ such that if $m(p) = q$ then $p \supset q$.*

**Proof** This follows from the existence of symmetric chain decomposition (see Greene and Kleitman (1976) for details).

A Symmetric Chain (SC) is a symmetric chain, that is, a chain

$$A_0 \subset A_1 \subset \cdots \subset A_t$$

in the Boolean lattice $\mathcal{B}_n$ whose ranks satisfy

$$|A_0| + |A_t| = n,$$

so the chain begins at rank $k$ and ends at rank $n - k$, increasing in size by one at each step.

A Symmetric Chain Decomposition (SCD) is a decomposition of $\mathcal{B}_n$, that is, a partition of the lattice into pairwise disjoint symmetric chains whose union contains every subset of $\{1, \ldots, n\}$

By definition, every SC can only include one point of any size, and any SC that includes a point of size $k$ also includes a point of size $n - k$. As an SCD provides disjoint cover of the hypercube, every point of size $k$ is part of a single chain. Each of chain contains only one point of size $n - k$, and as such any SCD defines a monotonic bijection from points of size $k$ to points of size $n - k$. ∎

### G.1.1. PROOF

Let $k \geq 0.5$ be the minimum recall rate. We will prove a stronger statement that for each $t \in [0, 0.5]$:

$$P(W_\rho^{g+} = t) \geq P(W_\rho^{g+} = 1 - t) \forall g \in \mathcal{G} \tag{15}$$

For individual datapoints, unless $t = \frac{s}{|\rho|}$ for some integer $s < |\rho|/2$, the equation trivially holds as left and right side of the equation are both 0.

When $t = \frac{s}{|\rho|}$, the above statement is equivalent to the probability of exactly $s \leq 0.5|\rho|$ members of the ensemble voting correctly is higher than the probability of exactly $s$ members voting incorrectly.

We will establish a bijective correspondence between each summand to a smaller summand in the expression

$$P(W_\rho^{g+} = 1 - t) = \sum_{\substack{p \in \rho \\ |q| = |\rho| - s}} I_q \tag{16}$$

By application of Lemma 2, followed by Lemma 1 we can rewrite:

$$P(W_\rho^{g+} = 1 - t) = \sum_{\substack{|q|=|\rho|-s \\ \forall s \le |\rho|/2}} I_q = \sum_{\substack{q=m(p) \\ |p|=s \\ \forall s \le |\rho|/2}} I_q \le \sum_{\substack{|p|=s \\ \forall s \le |\rho|/2}} I_p = P(W_\rho^{g+} = t) \forall g \in \mathcal{G} \quad (17)$$

as required.                                                                                 ∎

### G.2.  Minimum validation and evaluation sizes

**Statistical Framework:**  We can frame the problem of ensuring minimum recall as a one-sided hypothesis test:

$$H_0 : p_{\text{val}} = p_{\text{test}} = k \quad \text{vs.} \quad H_A : p_{\text{val}} > k. \quad (18)$$

Where $p_{\text{val}}$ is our threshold of interest. Because both the test set and validation sets are small, they both introduce sampling variability. Thus, we will explicitly account for the size of both.

The hypothesis-testing framework has a few assumptions. First, it assumes that the validation and test sets are *independently* drawn from the same distribution (an assumption we explicitly follow; see § 4). Second, it assumes that each positive instance is an independent **Bernoulli trial** that is either a true positive or a false negative. Finally, it assumes an approximately normal distribution. The normality assumption is met by the *Large Counts Condition*, which heuristically states that $\min(mk, m(1-k), nk, n(1-k)) \ge 10$, which in our case simplifies to $\min(\frac{m}{2}, \frac{n}{2}) \ge 10$. We thus minimally need roughly **20** positive instances for each group in both test and validation sets.

**Deriving minimums:**  Under $H_0$, the standard error of the difference between the minimum recall proportions in the validation and test set is:

$$\text{SE}_0 = \sqrt{k(1-k)\left(\tfrac{1}{m} + \tfrac{1}{n}\right)}.$$

The one-sided $z$ statistic fixing $p_{test} = k$ is

$$z = \frac{p_{\text{val}} - k}{\text{SE}_0}.$$

Requiring a significance level of $\alpha$ (i.e., $z \ge z_{1-\alpha}$) yields the minimal observable validation recall:

$$p_{\min} = k + z_{1-\alpha}\sqrt{k(1-k)\left(\tfrac{1}{m} + \tfrac{1}{n}\right)}.$$

For $k = 0.5$, this simplifies to the result in Eq. 5.

## G.3. Derivation of Equal Opportunity Bounds

We derive the fairness bounds for ensembles under approximate equal opportunity (or accuracy) constraints.

Starting from the definition of $k'$-approximate fairness for the ensemble, we have

$$k' = \max_{g \in \mathcal{G}} \mathbb{E}_{h \sim \rho}[L_g(h)](1 - \text{EIR}_{g^*}) - \min_{g \in \mathcal{G}} \mathbb{E}_{h \sim \rho}[L_g(h)](1 - \text{EIR}_{g^*}) \tag{19}$$

$$\leq \max_{g \in \mathcal{G}} \mathbb{E}_{h \sim \rho}[L_g(h)] - \min_{g \in \mathcal{G}} \mathbb{E}_{h \sim \rho}[L_g(h)](1 - \text{EIR}_{g^*}) \tag{20}$$

$$\leq k - \min_{g \in \mathcal{G}} \mathbb{E}_{h \sim \rho}[L_g(h)] \cdot (-\text{EIR})_{g^*} \tag{21}$$

$$\leq k + \max_{g \in \mathcal{G}} \mathbb{E}_{h \sim \rho}[L_g(h)]\text{DER}_{g^*} \tag{22}$$

where $g^*$ is an appropriate distribution (e.g., positives, negatives or all points) constrained to a particular group $g$. By substituting in the lower bound from Theorem 2 instead of 0, we obtain the slightly tighter bound of Equation 4.

## Appendix H. Detailed Related Work

**Fairness in Medical Imaging:** Deep learning-based computer vision methods have become highly popular for medical imaging applications (Cai et al., 2020), yet despite achieving near-human performance on top-level metrics (Liu et al., 2020), they consistently underperform for marginalised groups (Xu et al., 2024; Koçak et al., 2024). These biases persist across different domains and modalities from dermatology (Daneshjou et al., 2022) to chest X-rays (Seyyed-Kalantari et al., 2021) and retinal imaging (Coyner et al., 2023). For instance, there is pervasive bias in skin condition classification (Oguguo et al., 2023; Daneshjou et al., 2022; Groh et al., 2021), likely due to both bias in data collection (Drukker et al., 2023) and treatment procedures (Obermeyer et al., 2019).

Unfairness arise from different stages in the development process (Drukker et al., 2023). One persistent issue is unbalanced datasets (Larrazabal et al., 2020). Unbalanced datasets can lead to insufficient support for disadvantaged groups, which can lead to worse representations and more uncertain results (Ricci Lara et al., 2023; Mehta et al., 2024).

A successful approach to mitigating fairness is to do extensive hyperparameter and architecture search (Dutt et al., 2023; Dooley et al., 2022). By jointly optimising for fairness and performance, these methods can reduce the generalisation gap and outperform other methods. However, because of their computational cost, we do not compare against these in this work. However, our method can be built on top of the backbones found by the architecture search.

Defining fairness in the context of medical imaging is another challenge. While traditional fairness metrics, like equal opportunity (Hardt et al., 2016), are concerned with minimising disparities between groups, this might not be appropriate in a medical context. For instance, Zhang et al. (Zhang et al., 2022) find that methods which optimise this notion of group performance reduces the performance of all groups. This phenomenon of 'levelling down' (Zietlow et al., 2022) can have fatal consequences for patients and not meet the legal standards of fairness (Mittelstadt et al., 2024). Instead, researchers should strive to enforce minimum rate constraints, i.e., the performance of the worst-performing groups, which can help

reduce persistent problems of underdiagnosis and undertreatment of disadvantaged groups (Seyyed-Kalantari et al., 2021).

## Appendix I.  Alternative Backbones

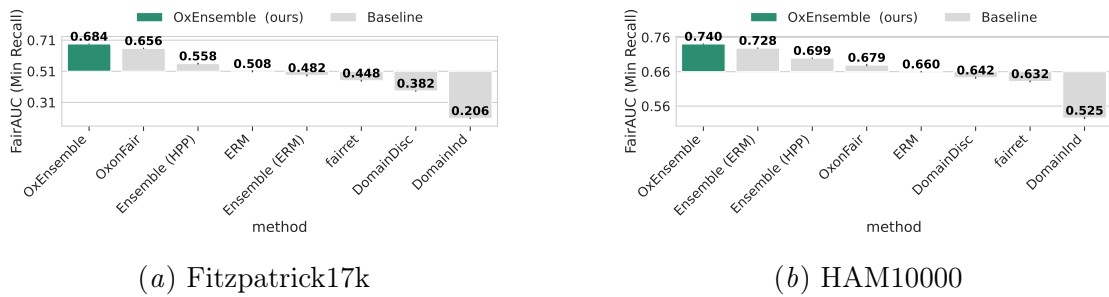

($a$) Fitzpatrick17k          ($b$) HAM10000

Figure 8: **Fairness–accuracy AUC (FairAUC) relative to ERM with Mobilenetv3 backbone**. OxENSEMBLE outperforms all baselines on Fitzpatrick17k (left) and HAM10000 (right). Error bars show 95% bootstrap CIs. Evaluation follows § 4 over minimum-recall thresholds in $[0.5, 1]$.

Here, we report the experiments conducted with a different backbone to show the robustness of our method. Specifically, we use the very small MobileNetv3 (Howard et al., 2019), which is popular for on-edge devices.

Figure 8 shows the main results. It shows that OxENSEMBLE convincingly outperforms all baselines on both HAM10000 and Fitzpatrick17k – the same results as for efficientnet in the main text.

