# OpenReview forum: "OxEnsemble: Fair Ensembles for Low-Data Classification"
_MIDL.io/2026/Conference — MIDL 2026 Poster_

### Official Review · Reviewer_P8xD · 2026-01-11

**Confidence:** 4
**Preliminary Rating:** 4
**Final Rating:** 5

**Summary:**

This paper proposes OxEnsemble, a method for training fair ensembles in low-data medical imaging settings. The approach uses a shared frozen backbone (EfficientNetV2) with multiple classifier heads trained on different data folds. Fairness constraints (like minimum recall) are enforced individually for each member using validation data. Predictions are aggregated via majority voting. Theoretical results extend "ensemble competence" to show that if members satisfy fairness constraints and are competent (recall > 0.5), the ensemble preserves these properties. Experiments on three datasets (HAM10000, Fitzpatrick17k, FairVLMed) show improved fairness-accuracy trade-offs compared to baselines.

**Strengths:**

- **Theoretical guarantees** -- The paper provides a rigorous extension of ensemble competence theory to fairness constraints, offering proofs for when ensembles preserve minimum rates and error parity. This moves beyond observational "ensembles usually help" claims.
- **Practical efficiency** -- The shared frozen backbone design is smart. It allows training a large ensemble (21 members) with inference cost comparable to a single model, addressing the main barrier to deployment.
- **Strong performance in low-data regimes** -- Results on Fitzpatrick17k (only 60 positives in minority group) are compelling, showing significant gains over ERM and other fairness methods.
- **Sample size guidance** -- Deriving the minimum validation recall ($p_{min}$) needed to guarantee test performance (Eq. 5) is a useful practical contribution for small datasets.
- **Clear focus on clinical reality** -- Arguments for "minimum recall" over "equal opportunity" in medical settings (avoiding leveling down) are well-reasoned and appropriate.

**Weaknesses:**

- **Independence assumption in theory** -- The theoretical proofs (e.g., G.1.1) rely on the assumption of independent errors between classifiers. In practice, ensemble members share the *same frozen backbone* and training data (folds overlap). This correlation violates the independence assumption, making the guarantees looser than stated. The paper acknowledges this briefly but relies on the result.
- **Dependency on frozen backbone** -- The method freezes the backbone to prevent overfitting. If the pretrained backbone (ImageNet) extracts biased features or misses features relevant to the minority group, the ensemble heads cannot recover them. The method assumes the backbone is sufficient, which may not hold for specialized medical tasks.
- **"Low-data" vs "Imbalanced"** -- The datasets have thousands of images total, just few positives for some groups. The method is more about *imbalanced* learning than true "low-data" (few-shot) learning. The distinction matters for positioning.
- **Baseline optimization** -- Line 459 states baselines use the "same configuration," implying they also use the frozen backbone. If so, does the performance gain come from the ensemble voting or just the regularization effect of the frozen backbone (preventing overfitting)? Ensuring the comparison isolates the *ensemble* benefit is crucial.

**Detailed Comments:**

- **Figure 4:** The Pareto frontiers are very effective at showing the trade-off.
- **Shared Backbone:** While efficient, using a single frozen backbone limits diversity. The "Disagreement-Error Ratio" (DER) suggests diversity helps. Have you analyzed the diversity (disagreement) of OxEnsemble members compared to a standard ensemble with separate backbones?
- **Eq 5:** This derivation for $p_{min}$ is useful. Is this specific to ensembles or applicable to any classifier validation? (It seems general).

**Justification Of Final Rating:**

Thank you for the detailed rebuttal. The clarifications have addressed my primary concerns regarding the independence assumption and the frozen backbone.

Specifically, confirming that *all* baselines used the same frozen backbone is crucial, as it isolates the performance gains to the ensemble method itself rather than the regularization effect of the backbone. I also appreciate the clarification that the independence assumption is strictly required only for the 0.5 recall threshold guarantee, while the broader competence framework holds without it.

The rebuttal successfully clarifies that the method's value lies in efficient data reuse—allowing fairness enforcement without sacrificing data to a held-out validation set—which is a valuable contribution for low-data medical imaging.

**Justification Of The Preliminary Rating:**

The paper tackles a relevant problem with a principled approach. The combination of theoretical guarantees (extending competence to fairness) and a practical, efficient architecture (shared backbone) is a strong contribution. Results on challenging datasets are convincing. Main concerns are the reliance on the independence assumption for proofs (which is violated by the shared backbone) and whether the gains come from the ensemble or simply using a frozen backbone that prevents overfitting in low-data regimes. Addressing the backbone's role in the rebuttal could raise this to a Strong Accept.

**Questions To Address In The Rebuttal:**

1. **Backbone bias:** Since the backbone is frozen and shared, how does OxEnsemble handle cases where the backbone itself encodes bias against the minority group? Can the heads overcome a fundamentally biased representation?
2. **Independence violation:** Your proofs assume independent errors. Given members share a backbone and data folds overlap, errors are definitely correlated. How robust are your guarantees to this correlation?
3. **Baseline fairness:** Did the baselines (e.g., DomainDisc, Fairret) also use a frozen EfficientNetV2 backbone? If not, is the performance difference due to overfitting in baselines vs. regularization in yours?
4. **Diversity:** How much disagreement (diversity) exists between heads on a frozen backbone? Is it sufficient to drive the ensemble gains predicted by the theory?
5. **Computational cost:** You mention training is ~3x slower than ERM. Is this for the full 21-member ensemble?

---

> ### Author Response · Authors · 2026-01-20
> **Addressing backbone, independence, baselines, diversity etc.**
>
> Thank you for your helpful comments, particularly regarding strengthening our positioning. We are glad you acknowledge our strong theoretical contributions as well as our convincing empirical results. We hope that our clarifications of the backbone can increase your score.
>
> ## Theory
> > Independence assumption in theory.
> > Independence violation: Your proofs assume independent errors. [...] How robust are your guarantees to this correlation?
> > How robust are your guarantees to this correlation?
>
> The independence assumption is only used for one theoretical result -- that a minimum recall rate threshold of 0.5 guarantees competence. For all other results, independence is not required.
>
> Even for the minimum recall rate, we know that without assuming independence, some recall rate will guarantee competence (see section 3.2.2), but we would need to determine this rate empirically, and empirically 0.5 (see figure 2) seems to be the threshold where it starts occuring.
>
> Crucially, we also empirically show that our method does indeed convincingly outperform existing baselines.
>
> > Eq. 5: This derivation for minimum validation recall is useful. Is this specific to ensembles or applicable to any classifier validation?
>
> This is generally applicable to any system where the minimum recall can be enforced on the validation set. We will emphasise this in the text.
>
> ## Backbone
> >  If the pretrained backbone (ImageNet) extracts biased features or misses features relevant to the minority group, the ensemble heads cannot recover them. The method assumes the backbone is sufficient, which may not hold for specialized medical tasks.
>
> This is really a situation where *all* options suck. With the limited amount of data available, we don't expect training from scratch to work. Updating the entire backbone also allows new feature biases to be introduced, and existing ones to be amplified, and shouldn't be seen as a panacea. It also makes overfitting more likely and requires much more care in early stopping. If a pretrained backbone has already suffered from feature collapse, there is no guarantee that fine-tuning will recover it.
>
> In general, the approach of Oxonfair, which is inherited by our ensembles, is that you should train as performant a classifier as possible, and fix relevant biases by merging multiple heads. In this context of limited data, using a frozen backbone is the go-to option for maximizing performance, with other options typically being explored for completeness.
>
> That's not to say that using a frozen backbone is always the best option, just that it is generally a safe and sensible design decision.
>
> > Have you analyzed the diversity (disagreement) of OxEnsemble members compared to a standard ensemble with separate backbones?
>
> We have not. It's worth mentioning that if the backbone is pretrained, there will still be a lot of correlation between the different backbones after fine-tuning.
>
> > Diversity: How much disagreement (diversity) exists between heads on a frozen backbone? Is it sufficient to drive the ensemble gains predicted by the theory?
>
> The theory bounds the performance gains as a function of the diversity.
>
> > If so, does the performance gain come from the ensemble voting or just the regularization effect of the frozen backbone (preventing overfitting)?
>
> The performance gain comes from efficient data reuse, which means fairness can be enforced without using a held-out validation set to accurately estimate error rates.
>
> A small boost in performance due to use of ensembles is generally expected but not needed for the method to be worth using.
>
> > Did the baselines (e.g., DomainDisc, Fairret) also use a frozen EfficientNetV2 backbone?
>
> All the baselines used the same frozen backbone. We kept the backbone the same to isolate the effects of our proposed method. Specifically, we compare against both a standard ensemble trained with ERM and ensemble variations from existing literature. By keeping processing techniques identical (also in terms of, e.g., rebalancing), we can be confident that the improvements we see come from our proposed method.
>
> ## Clarification
> > "Low-data" vs "Imbalanced"
>
> **Action N: We will clarify the distinction between low-data and imbalanced data.**
>
> The key issue here is *a lack of data* for underrepresented groups. If we kept the data ratios the same (i.e. just as imbalanced) but increased the amount of data by 1000x, the algorithm would be likely be much less unfair.
>
> > Computational cost: You mention training is ~3x slower than ERM. Is this for the full 21-member ensemble?
>
> Yes, it is for the full 21-member ensemble. We show slightly slower training and equivalent inference cost in Appendix F.
>
> **Thank you again for taking the time to review the paper and providing helpful feedback! Do the above actions address your concerns with the paper? If not, what further clarification or modifications could we make to improve your score?**

---

### Official Review · Reviewer_TXPP · 2026-01-12

**Confidence:** 3
**Preliminary Rating:** 4

**Summary:**

The paper proposes OxEnsemble, a theoretically grounded approach for creating fair ensembles. Ensemble approaches are suitable in low-data settings, but understanding of ensemble fairness in such scenarios unclear. The authors extend existing theoretical results about ensemble classifier to fairness & show enforcing fairness constraints at the member level and provides theoretical guarantees that these constraints are preserved (and often improved) at the ensemble level via majority voting.

The authors evaluate OxEnsemble on three medical imaging datasets (HAM10000, Fitzpatrick17k, and Harvard-FairVLMed) show that it achieves very similar accuracy-fairness trade-off as strong baselines like Fairret and HPP. The paper is well-written, technically sound, and addresses an important problem.

**Strengths:**

- The paper addresses an important problem: how to enforce fairness when data for minority groups is extremely scarce—a common problem in medical imaging. The paper is generally written.
- The most significant contribution is the theoretical characterization of when ensembles improve fairness. The idea, even though seems like a direct extension of extension of the concept of ensemble competence (Theisen et al., 2023), it is a novel application.

On Experiments.
 - The choice of fairness metrics & baseline seem reasonable. The focus on minimum rate constraints (ensuring a floor for the worst-performing group) is appropriate for safety-critical domains.
- I like that authors report AUC & pareto frontiers, which provide a more complete picture of the accuracy-fairness trade-off.
- Using a shared backbone with multiple heads allows the ensemble to be trained and run with a marginal increase in compute over a single model, making it a computationally efficient choice.

**Weaknesses:**

- Some assumptions made in theory may not be satisfied in practice. For example, the theoretical guarantees rely on the assumption that ensemble members make independent errors. In practice, because all members share a pre-trained backbone (EfficientNetV2), their errors are likely to be correlated. This should be discussed more.
- The paper relies on "OxonFair’s multi-head surgery" to combine heads. For a self-contained research paper, a slightly more detailed explanation of how this surgery handles the trade-off during training would help the reader understand the approach better.
- Author majorly use minimum rate constraint as fairness measure. While this is interesting, it would also be interesting to see if the results generalize to other measures such as equality of odds/opportunity.

**Detailed Comments:**

- Please add a self-contained description of the approach. Mainly I would like to see how the fairness constraints are achieved by combining the head.

- It would be interesting to see if the ensembled approaches work when the if the backbone were not frozen. More interestingly it would be interesting to see if the fairness can be preserved with diverse backbones.

- As mentioned in weakness, it would be interesting to see  if OxEnsembled can be easily extended to other fairness measures.

**Justification Of The Preliminary Rating:**

The paper makes interesting theoretical observations & validates those empirically. Even though the empirical approach doesn't beat baselines clearly, I think its an interesting work. The empirical results are very similar to baselines while being theoretically solid.

**Questions To Address In The Rebuttal:**

See comments & weaknesses.

---

> ### Author Response · Authors · 2026-01-20
> **Assumptions, additional experiments, and clarifications**
>
> Thank you for your helpful comments, particularly regarding making our contributions clearer. We are glad you acknowledge the contribution of our theory and our strong empirical evaluation.
>
> ## Theory clarification
> > [...] In particular, the guarantees rely on ensemble members making independent errors [...]
>
> The independence assumption is only used for one theoretical result - that a minimum recall rate threshold of 0.5 guarantees competence. For all other results, independence is not required.
>
> Even for the minimum recall rate, we know that without assuming independence, some recall rate will guarantee competence (see Section 3.2.2), but we would need to determine this rate empirically, and empirically 0.5 (see figure 2) seems to be the threshold where it starts occurring.
>
> Crucially, we also empirically show that our method does indeed convincingly outperform existing baselines.
>
> ## Fairness metrics
> >  [...] It is unclear whether the results generalize to other fairness notions [...]
>
> We explicitly study equality of opportunity, as well as minimum recall constraints, both theoretically (through bounding improvements; see section 3.2.1 (2)) as well as empirically through the FairVLMed dataset.
>
> The multi-head surgery can be used to enforce arbitrary group fairness constraints, so in principle it can be applied to any other fairness definition, albeit without theoretical guarantees.
>
> ## Clarifications & Ablations
>  > The paper relies on “OxonFair’s multi-head surgery” to combine heads, but a more self-contained explanation would help [...]
>
> **Action 1: We will add a self-contained explanation.**
>
> Thank you for the suggestion. We will add a self-contained explanation of the multi-head surgery.
>
> > It would be interesting to see if the ensembled approaches work when the if the backbone were not frozen.
>
> **Action 2: We will add experiments with a different backbone.**
>
> Unfreezing the backbone would hurt the efficiency of training and potentially introduce more dependence between members.
>
> We will add an experiment with MobileNetV3 to show that our method works for multiple backbones. Qualitatively, we find very similar results; OxEnsemble is still the best performing method across the datasets.
>
> **Thank you again for taking the time to review the paper and providing helpful feedback! Do the above actions address your concerns with the paper? If not, what further clarification or modifications could we make to improve your score?**

---

### Official Review · Reviewer_AZV6 · 2026-01-15

**Confidence:** 3
**Preliminary Rating:** 4
**Final Rating:** 5

**Summary:**

The paper addresses the fairness issue in medical imaging classification under limited data settings by proposing OxEnsemble. OxEnsemble is a single-encoder multi-head structure, where the outputs from these heads are aggregated by majority vote. The paper presents the theoretical proof of the fairness guidance, and runs experiments on three medical datasets.

**Strengths:**

1. The paper is well-motivated to address critical fairness problems under limited data settings, especially focusing on false negatives.
2. The architecture of OxEnsemble is clearly presented and the mathematical proof of fairness constraints makes sense.
3. The experiments are performed across three medical datasets, and compares OxEnsemble with multiple previous ensemble methods, demonstrating the better capability of OxEnsemble by achieving better FairAUC.

**Weaknesses:**

1. It is said “For baselines without explicit thresholding, we generate Pareto frontiers by varying global thresholds on validation data.”, I think the author assumes the same distribution between the validation and test data, what if in practice this is not? Adding a discussion may strengthen the paper.
2. The paper only evaluates the performance on binary classification tasks. However, the effectiveness of multi-class tasks remains unclear.
3. The rationale of threshold selection needs clarification.

**Detailed Comments:**

See Strengths and Weaknesses

**Justification Of Final Rating:**

I appreciate the author's response and clarifications on threshold selection and metric design. Even though multi-class fairness remains challenging, the author clearly justify their choices. I am glad to increase the score.

**Justification Of The Preliminary Rating:**

The paper proposes OxEnsemble with good motivations, followed by clear theory proof and comprehensive experiments. However, adding the clarification of sensitivity of the threshold selections and adding multi-class tasks can further improve the paper.

**Questions To Address In The Rebuttal:**

Clearly discuss the rationale behind the threshold value selection and sensitivity. Also, adding multi-class tasks may strengthen the paper.

---

> ### Author Response · Authors · 2026-01-20
> **Clarifying thresholds and metrics**
>
> Thank you for your helpful comments, particularly regarding clarifying the thresholding approach. We are glad you acknowledge the superiority of our method and the clarity of our contributions.
>
> ## Thresholds & Distribution shifts
> >  It is said “For baselines without explicit thresholding, we generate Pareto frontiers by varying global thresholds on validation data.” The author assumes the same distribution between the validation and test data.
>
> This text refers only to baseline methods (not OxonFair, and not our ensembles) that do not support minimum recall constraints. In fact, our ensembles are more robust (careful recycling allows us to set minimum recall constraints by using all the data in training rather than a smaller validation set). The question as to how methods not proposed by us, and not used by our ensembles, cope with out-of-distribution data is out of scope for this work.
>
>
> > The rationale of threshold selection needs clarification.
>
> **Action 1.2: We will further clarify threshold selection.**
>
> As we argue in section 4, the choice of threshold/ minimum recall rate is a deployment decision. We therefore report FairAUC which corresponds to the average performance over a range of thresholds. We will further clarify both how thresholds are fitted in OxEnsemble through a self-contained description of OxonFair's multi-head surgery and how we tune the thresholds for the baselines.
>
> ## Multi-class classification
> > [...] the effectiveness on multi-class tasks remains unclear.
>
> Yes, this reflects the lack of literature on multiclass fairness. In particular, there is an almost complete absence of relevant metrics. Demographic parity has received some attention in terms of multi-class extensions, but it and its multiclass extension implicitly assume that the distribution of illness should not vary with the protected attribute (e.g. ethnicity or gender). This is inappropriate for most illness where relevant environmental and genetic factors can vary with the protected attribute, and drive different illness prevalences.
>
>
> **Thank you again for taking the time to review the paper and providing helpful feedback! Do the above actions address your concerns with the paper? If not, what further clarification or modifications could we make to improve your score?**

---

### Official Review · Reviewer_Zbdc · 2026-01-15

**Confidence:** 3
**Preliminary Rating:** 3

**Summary:**

In this paper, the authors propose OxEnsemble, an efficient ensemble training strategy that enforces fairness in low-data settings. The method addresses unfair classification arising from limited and imbalanced datasets, leading to more consistent outcomes and improved fairness–accuracy trade-offs across multiple challenging medical datasets.

**Strengths:**

- The paper is clearly structured, well motivated, and well written
- Models are trained to satisfy specific fairness constraints, and their predictions are aggregated, potentially leading to more robust predictions
- Efficient ensembling is achieved by sharing a common backbone across models and concatenating classifier head across ensemble members, enabling better use of scarce data while leveraging disagreement for robustness.
- The approach is evaluated against several baselines, showing that it achieves better accuracy with lower fairness violation

**Weaknesses:**

- Training ensemble members on different data folds can lead to substantial disagreement, particularly when majority voting is used, resulting in highly variable predictions across models. It is unclear how the authors address this issue
- It is unclear how sensitive the results are to the choice of backbone CNN; experiments with additional backbones could help assess the generalizability of the findings.

**Detailed Comments:**

- Computational efficiency of the ensemble: It is unclear how the proposed ensemble remains computationally efficient, particularly given the need to train and maintain multiple heads
- Diversity among ensemble members: The paper does not clearly explain how diversity is enforced across ensemble members, especially to ensure that each model learns complementary information from underrepresented samples rather than converging to similar decision boundaries
- Fairness in small vs. large datasets: The method is designed on relatively small imaging datasets where minority groups are underrepresented. It remains unclear how the approach would scale to larger datasets, and what aspects of the proposed framework inherently constrain it to the low-data regime
- Clarity of Figure 1(a): The columns in Fig. 1(a) are not sufficiently explained, making it difficult to interpret the illustrated workflow
- Hyperparameter selection: Since each ensemble member is trained on a different data fold, the strategy for hyperparameter selection and tuning is unclear.

**Justification Of The Preliminary Rating:**

The authors address an important problem in medical diagnosis under real-world constraints, including group imbalance and data scarcity. They propose a framework that constructs efficient ensembles of fair classifiers designed to enforce fairness in these challenging settings. The method is evaluated on multiple datasets and compared against several baseline approaches, demonstrating the superiority of the proposed approach.

**Questions To Address In The Rebuttal:**

See Weaknesses and Detailed Comments

---

> ### Author Response · Authors · 2026-01-20
> **Clarifications and additional experiments**
>
> Thank you for your helpful comments, particularly regarding improving the clarity of our contribution. We are glad you acknowledge the importance of the problem we tackle and the rigour of our evaluation.
>
> ## Diversity & Training
> > Training ensemble members on different data folds can lead to substantial disagreement [...]
>
> As described in section 3.2, disagreement is a key feature of ensembles. Providing the theoretical requirements set out in eq. 1 hold, there are no concerns.
> For instances with high disagreement, we expect ensembles to outperform ERM as shown in equation 2.
>
> > The paper does not clearly explain how diversity is enforced across ensemble members [...]
>
> Diversity is not enforced, but simply occurs as a consequence of training an ensemble on different subsets of the data (see section 3.1).
> Unlike standard ensembles, we are looking for a data-efficient way to reuse held-out validation data used to enforce fairness. We do not rely on diversity, but want a way to ensure fairness, even in the presence of diversity.  We show empirically and theoretically that our proposed ensemble does this.
>
>
> > [...] the strategy for hyperparameter selection and tuning is unclear.
>
> All heads share the same hyperparameters which are set in appendix A.
>
> There are few sensitive hyperparameters in our method. The main additional parameter is the size of the ensemble, which we set to a relatively large value based on theoretical arguments. Thus, OxEnsemble doesn't need a clean held-out validation set to tune hyperparameters.
>
>
> ## Backbone
> > It is unclear how sensitive the results are to the choice of backbone [...]
>
> **Action 1: We will add results with a different backbone.**
>
> We add results to the appendix using MobileNetv3 with the same procedure. The results are qualitatively similar; our OxEnsemble method still clearly outperform all baselines.
>
> >  It is unclear how the proposed ensemble remains computationally efficient [...]
>
> As described on page 4, the computational benefits come from sharing a common, frozen backbone. This means that we don't need to train $M$ individual classifiers and inference can be performed with a single forward pass. We empirically analyse efficiency in Appendix F, which shows comparable efficiency to training ERM.
>
> All additional heads can be contained in a single linear layer of the network, of size (2x number of heads x size of backbone output). This corresponds to k parameters vs the (big number) of parameters in the backbone.
>
> ## Clarity & Contribution
> > Clarity of Figure 1(a)
>
> **Action 2: We will add descriptions of the criteria.**
>
> We add descriptions of the columns we compare against in the caption and related works section. This should improve the clarity of the comparison table.
>
> > Fairness in small vs. large datasets [...]
>
> We tackle the problem of small group datasets since existing fairness methods performs poorly as evidenced by Zong et al. (2022) and our empirical evaluations. For large group-level datasets, we expect that our method would still perform well - but so would other methods such as OxonFair - see their original paper. We therefore focus our evaluations on the domains where our contribution is the strongest.
>
> **Thank you again for taking the time to review the paper and providing helpful feedback! Do the above actions address your concerns with the paper? If not, what further clarification or modifications could we make to improve your score?**

---

### Author Rebuttal · Authors · 2026-01-22

**Rebuttal:**

We thank all the reviewers for their helpful comments that recognised our state-of-the-art results and novel theory. Given the tight turnaround, the amended submission should be seen as a first pass to incorporate some of their comments, and not the final version.

Again, we would like to thank everyone involved for their time.

**Supporting Material:**

/attachment/2d6b8a93b00f1a66c1d3484a3e5928a1c50c8dc5.pdf

---

### Meta-Review · Area_Chair_DMBW · 2026-02-08

**Recommendation:** Accept (Oral)
**Confidence:** 4

**Metareview:**

All reviewers appreciated the paper's methodology and evaluation. The paper had 3 weak accept and 1 borderline before the rebuttal. Two reviewers updated their rating to strong accept after the rebuttal, while the other two reviewers didn't respond (and as such, didn't update their rating) after the rebuttal. Considering everything, I am recommending acceptance of the paper.

---

### Decision · Program_Chairs · 2026-02-13

Accept (Poster)